# Effect of Humidity-Triggered Controlled-Release 1-Methylcyclopropene (1-MCP) on Postharvest Quality of Papaya Fruit

Chang Shu [1,2], Marisa M. Wall [1], Peter A. Follett [1], Nobuko Sugimoto [3], Jinhe Bai [4] and Xiuxiu Sun [1,*]

[1] United States Department of Agriculture, Agricultural Research Service, Daniel K. Inouye U.S. Pacific Basin Agricultural Research Center, 64 Nowelo Street, Hilo, HI 96720, USA; chang.shu@usda.gov (C.S.); marisa.wall@usda.gov (M.M.W.); peter.follett@usda.gov (P.A.F.)

[2] Oak Ridge Institute for Science and Education, 1299 Bethel Valley Road, Oak Ridge, TN 37830, USA

[3] Verdant Technologies, 12600 East Arapahoe Road, Suite 175, Centennial, CO 80112, USA; nobuko.sugimoto@verdant-tech.com

[4] United States Department of Agriculture, Agricultural Research Service, U.S. Horticultural Research Laboratory, 2001 South Rock Road, Fort Pierce, FL 34945, USA; jinhe.bai@usda.gov

\* Correspondence: xiuxiu.sun@usda.gov; Tel.: +1-808-959-4307

**Abstract:** Papaya (*Carica papaya* L.) is a valuable economic crop that is widely cultivated in tropical and subtropical regions but has a short storage and shelf life. Exploring effective strategies to improve the postharvest quality of papaya is important. This study explored the effect of humidity-triggered controlled-release 1-methylcyclopropene (1-MCP) sheets on the postharvest quality of papaya fruit. 'Rainbow' papayas underwent cold storage at $10 \pm 0.5$ °C, RH 85% $\pm$ 2% for 14 days, and then were transferred to $20 \pm 0.5$ °C, RH 85% $\pm$ 2% for 10 days to simulate shelf life. The 1-MCP sheets were cut into different sizes and placed in storage containers in advance to create corresponding concentrations at 0.5, 1.0, 2.0, and 4.0 ppm. Results showed that 1-MCP treatment inhibited fruit softening, and reduced weight loss and peel color deterioration without causing any physiological disorders. The 1.0–2.0 ppm 1-MCP-treated fruit received the highest score for papaya flavor and sweetness respectively and the lowest score for off-flavor. The humidity-triggered controlled-release 1-MCP sheets are effective and convenient, and they can serve as an important tool for regulating postharvest papaya ripening with economic benefits.

**Keywords:** papaya; 1-methylcyclopropene (1-MCP); postharvest quality; ripening

## 1. Introduction

Papayas (*Papaya carica* L.) are a rich source of vitamins C and A, and are widely consumed in Hawaii and the Pacific Basin [1]. 'Rainbow' papaya is the most popular variety produced in Hawaii, accounting for about 70% of the papaya acreage [2]. Hawaii's production of 'Rainbow' papayas was 13.4 million pounds in 2021, which was a 62% increase from 2020 and was the state's highest output since 2017. Papayas are climacteric fruit that ripen and soften quickly, leading to considerable postharvest losses during transport from tropical production areas to consumption areas on the US mainland, Canada, and Japan. Additionally, they are chilling-sensitive, and the lowest recommended temperatures for storage are 7 to 13 °C [3], contributing to high postharvest metabolism and short shelf life. Therefore, developing effective postharvest strategies is of economic significance.

1-Methylcyclopropene (1-MCP), a synthetic plant growth regulator, is used commercially to slow the ripening of fresh fruit by blocking ethylene receptors [4]. 1-MCP is structurally similar to the natural plant hormone ethylene, and it has been widely applied for postharvest fruit preservation. For example, 1-MCP has been reported to effectively slow ethylene release, delay softening, reduce malondialdehyde content, and inhibit the activity of cell-wall-degrading enzymes in mangoes [5]. The application of 1-MCP reduced





softening in apple fruit by an average of 12% and delayed the ripening process by lowering the rate of sucrose hydrolysis [6]. Postharvest treatment of pear fruit with 1-MCP induced multiple changes in the bacterial and fungal microbiota on the surface of the fruit, resulting in significantly reduced disease incidence [7]. However, papayas and many other fruits reach an edible stage only after fruit has ripened/softened, and inappropriate 1-MCP treatment often makes papayas [8,9] and pears [10] lose ripening capacity. Fortunately, 1-MCP treatment of more mature fruit delays its ripening, without completely blocking it, such as in the case of 1-MCP treatment applied to partially ripened pears [11] or to higher-maturity (such as half-yellow harvest) papayas [9]. However, even for ripening triggered fruit, the 1-MCP effects differ depending on the maturity stage, dosage, and application conditions (temperature and humidity).

Previous studies already applied 1-MCP on postharvest papaya storage, and arrived at 1.0 to 1.5 ppm as the recommended dose [12–14]. However, traditional 1-MCP is not convenient to use, and too high doses can cause physiological disorder. Novel forms of 1-MCP need to be further explored for convenient application in industry. The novelty of this study is the application of a new technology developed by Verdant Technologies, so-called HarvestHold®, for papaya postharvest storage. This is a new application of 1-MCP, which involves clathrates of 1-MCP with $\alpha$-cyclodextrin. Certain humidity and temperature conditions will trigger the release of 1-MCP. The mean clathrate size is 3–15 μm; this maintains the 1-MCP release rate at a relatively slow level. The clathrates are printed on the film so that they can be easily applied [15]. The traditional 1-MCP formulation is encapsulated in $\alpha$-cyclodextrin or other wall materials and is dissolved in water to release gaseous 1-MCP to fumigate the product. Therefore, it is a high-dose single treatment, which requires a long treatment time and fumigation chambers [16]. HarvestHold® enables the product to stay in the 1-MCP atmosphere when compared with the classic 1-MCP solutions, to extend the usable product life of fruits, vegetables, and flowers. Along with its storability and portability, HarvestHold® is a promising strategy in the postharvest industry.

Ethylene-triggered fast ripening and senescence significantly reduce postharvest papaya commodity values. In the present study, we tested the effects of different concentrations of controlled-release 1-MCP on the quality of papayas harvested at half-yellow skin color. The objective of this research was to apply commercially available humidity triggered controlled-release 1-MCP sheets to the Hawaii papaya cultivar and find the best concentration to extend its postharvest shelf life and quality.

## 2. Materials and Methods

### 2.1. Fruit and Treatments

'Rainbow' papayas were harvested at half color stage (50% of the surface was yellow) in Hilo, Hawaii in November 2022. Fruits were uniform in size (about 0.9 kg/fruit), and free of any blemishes or deformities. Controlled-release 1-methylcyclopropene (1-MCP) sheets (HarvestHold Fresh®; $25 \times 33$ cm$^2$) were obtained from Verdant™ Technologies (Centennial, CO, USA). The sheets comprised a composition consisting of 1-methylcyclopropene clathrate and $\alpha$-cyclodextrin (1-MCP@$\alpha$-cyclodextrin) with a mean particle size between 3 μm and 15 μm.

Four 1-MCP treatments were achieved by applying different sizes of 1-MCP sheets: 1 sheet, 1/2 sheet, 1/4 sheet, and 1/8 sheet which roughly corresponded to calculated concentrations of 4.0, 2.0, 1.0, and 0.5 ppm, respectively. This was calculated on the basis of the size of the commercial container used in this study, the volume of the fruit, and the headspace volume over the fruit, according to the manufacturer's guidance. Fruit without 1-MCP sheets were held as a control. The five treatments were replicated three times, and, for each replicate, 90 fruits were divided among five reusable plastic containers (RPCs; 40 cm $\times$ 60 cm $\times$ 20 cm). In each RPC, a 1-MCP sheet was placed on top of the fruit (Figure 1), and the five boxes of fruit were stacked and capped with a piece of cardboard to mimic shipping conditions. After inserting the 1-MCP sheet, fruit in RPCs were stored at $10 \pm 0.5$ °C and $85\% \pm 2\%$ RH for 14 days to simulate cold storage and

marine transportation, and then transferred to 20 ± 0.5 °C and 85% ± 2% RH for 10 days to simulate commercial retail conditions. Fruits were taken out after intervals of cold storage and room temperature (RT) storage (on days 7, 14, 14 + 2, 14 + 4, 14 + 6, 14 + 8, and 14 + 10 [cold storage + RT]) to measure weight, firmness, color, total soluble solids content (SSC), titratable acidity (TA), and decay rate. Three of the five RPCs in each replicate were used for visually estimating decay and weight loss on each sampling date. The other two boxes were used for destructive sampling for firmness, SSC, TA, and color. Sensory evaluations were conducted on remaining fruit after 14 + 10 days storage at the end of the experiment.

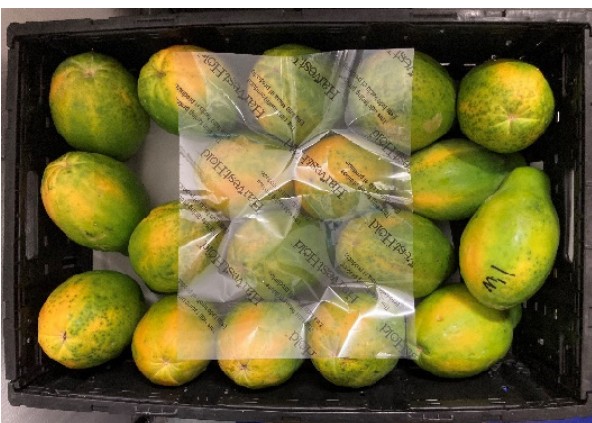

**Figure 1.** Controlled-release 1-methylcyclopropene (1-MCP) sheet and its application to papayas in a reusable plastic container.

### 2.2. Color

Fruit surface color and pulp color were measured on three papayas at four points near the equator using a Minolta chromameter (model CR-300, Minolta Corp., Ramsey, NJ, USA) and recorded as CIE L*, a*, and b*, where L* indicates lightness read from 0 (black) to 100 (white). The positive a* value indicates red color while the negative a* value represents green color. Similarly, positive and negative b* values indicate yellow and blue color, respectively.

### 2.3. Weight Loss

Weight loss (%) was calculated as the percentage of weight lost per original fruit weight in each container. In each replicate, there were three containers with 18 fruit each, and the weight of each container was measured. Linear regressions were established during cold storage and shelf life to represent the daily weight loss of the fruit.

### 2.4. Firmness

Fruit firmness was measured by a texture analyzer (Model Chatillon LTCM-100, AMETEK, Inc., Berwyn, PA, USA) using a 0.6 cm diameter of probe to puncture the fruit without peel to a depth of 1 cm at a speed of 25.4 cm min$^{-1}$ at the equatorial region. The data were expressed in Newtons (N). In each replicate for each time point, a total of three fruits were measured at two randomly selected locations near the fruit's equator.

### 2.5. Total Soluble Solid Content and Titratable Acidity

Fresh pulp (20.0 g) from three fruit per replicate was ground, and the juice was extracted by squeezing through two layers of muslin cloth. The total soluble solid content (SSC) was determined by a digital refractometer (PAL-3, ATAGO U.S.A., Inc., Bellevue, WA, USA), and reported as a percentage. The titratable acidity (TA) was determined by an acidity meter (GMK-835F, ATAGO®, ATAGO U.S.A., Inc., Bellevue, WA, USA), which measures the total amount of hydrogen ions and expressed as the percentage TA [17].

### 2.6. Decay Rate

The fruit with visible lesions were defined as decayed, and the decay rate was expressed as the percentage decayed fruit from the total fruit in each box.

### 2.7. Sensory Evaluation

Eating quality of fruit pulp was evaluated by a ten-member trained sensory evaluation panel. Panelists were presented with slices from randomly selected fruit from the last day of the experiment (14 + 10). Sweetness, sourness, papaya flavor, off-flavor, firmness, and overall eating quality were rated by sensory panelists using a 0–10 (low to high) hedonic scale.

### 2.8. Statistical Analysis

Data were organized and graphed using Excel (Microsoft Corp., Seattle, WA, USA), then analyzed using JMP statistical analysis software (version 16; SAS Institute, Cary, NC, USA). Analysis of variance (ANOVA) was used to evaluate the effect of treatments on papaya quality attributes. For significant treatment effects, means separations were applied using Tukey's HSD test at $\alpha = 0.05$ within different groups at the same timepoint. At least three replications were conducted for all experiments to provide the papaya response data.

## 3. Results and Discussion

### 3.1. Fruit Appearance and Color Change

As shown in Figure 2A, in the control group, part of the epidermis appeared yellow after 14 days of cold storage, while all the 1-MCP groups retained a green-colored peel during cold storage. Overall, the peel color of the control group turned yellow rapidly, while it was slower in the 1-MCP groups. At the final timepoint (day 14 +10), the peel of the control group and 0.5 ppm group showed significant orange color, while the 2.0 ppm and 4.0 ppm 1-MCP groups still maintained some green area. In contrast, the pulp color was not different between any treatment throughout the experiment (Figure 2B). Notably, after a week at room temperature, the pulp of the control group softened significantly and became translucent, while the pulp of all the 1-MCP groups was opaque and exhibited normal orange color.

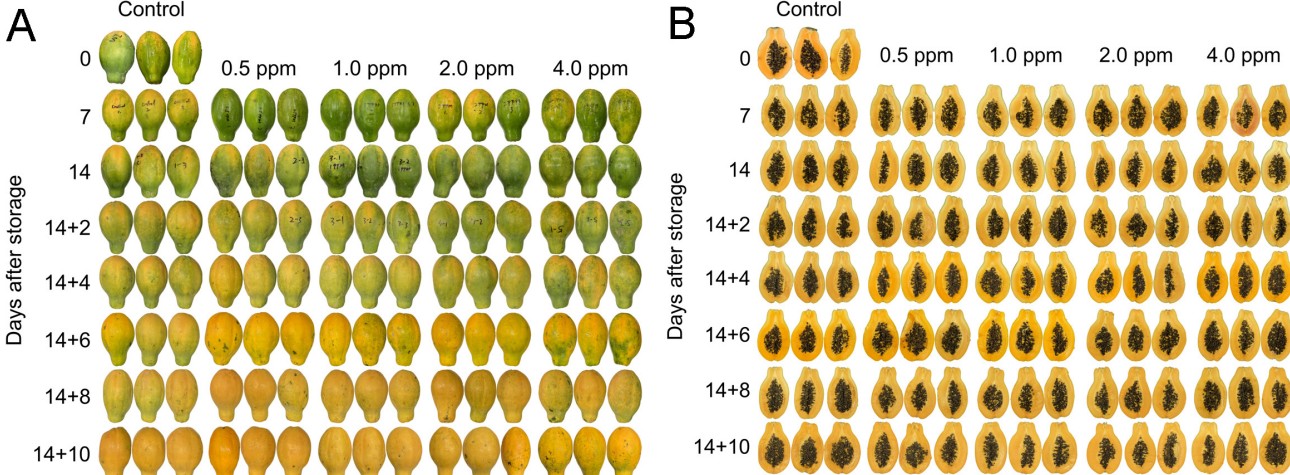

**Figure 2.** Effect of 1-MCP treatment on papaya appearance change during storage. Appearance of peel (**A**) and pulp (**B**). Papaya fruits were treated by 0 (control), 0.5, 1.0, 2.0, and 4.0 ppm 1-MCP, and the fruits were stored at $10 \pm 0.5$ °C and relative humidity (RH) 85% ± 2% for 14 days, and then transferred to $20 \pm 0.5$ °C and RH 85% ± 2% for 10 days.

The L* value represents the brightness of the peel, and the L* value increased continuously during storage (Table 1). During cold storage, the L* value of the control group

was the highest, followed by the 0.5 ppm 1-MCP group. At room temperature, the L* value of each group gradually increased over time. The L* values of the control, 0.5, and 1.0 ppm 1-MCP groups increased significantly, while its values for the 2.0 and 4.0 ppm 1-MCP groups were the lowest on day 14 + 2 and 14 + 6. However, at the final timepoint, the differences between all the groups were not significant. The a* value of all groups generally increased over time (Table 1), indicating that peel color was turning from green to red. There was no significant difference between the 4.0 ppm group and the other 1-MCP groups during cold storage and the first 4 days at room temperature, but the 4.0 ppm group was the lowest after 6 to 10 days at room temperature.

On day 14 + 10, a* values of the control, 0.5 ppm, 1 ppm, 2 ppm, and 4.0 ppm groups were 22.28, 24.44, 17.19, 15.81, and 4.61, respectively. This indicated that 1-MCP@$\alpha$-cyclodextrin clathrate effectively slowed down the chlorophyll degradation of the peel, when compared to control fruit. The increase in b* indicates the continuous yellowing of the peel (Table 1). The b* value of the control group was the highest during cold storage and the first 4 days at room temperature. On day 14 + 6 and 14 + 10, the b* value of all groups approached a similar level, and the difference was not significant. This indicated that all of the 1-MCP@$\alpha$-cyclodextrin clathrates effectively slowed the yellowing of the peel during cold storage and the first 4 days of shelf life, and the slowing of color change was dose-dependent and greatest in the 4.0 ppm treatment.

Generally, the color of fruit is related to changes in the fruit's composition. Low-maturity papaya peel is green due to its high chlorophyll content, which rapidly declines as the fruit ripens. Along with the accumulation of lutein and β-carotene, the fruit peel shifts from green to yellow [18]. Application of 1-MCP delays the peel color change, which is related to its inhibition of chlorophyll catabolic gene expression to suppress chlorophyll degradation, which significantly maintains the green color of apple [19] and broccoli [20]. The wax layer is intact in low-maturity fruit, in which the fruit surface is dense, leading to a higher L* value. As the fruit matures and deteriorates, the composition of the wax changes and the skin color becomes darker. Additionally, phenolic compounds in pulp cause browning, causing the L* value to decrease continually. A study with papayas also found that, with ripening, peel L* increased gradually [21]. The yellowing of the papaya skin is due to the degradation of chlorophyll, accompanied by the synthesis of carotenoids, which ultimately appear yellow/orange [22].

**Table 1.** Effect of 1-MCP treatment on papaya peel color, pulp firmness, SSC, and total acid content *.

| Parameters | Group | Storage Time (Day) | | | | | | | |
|---|---|---|---|---|---|---|---|---|---|
| | | 0 | 7 | 14 | 14 + 2 | 14 + 4 | 14 + 6 | 14 + 8 | 14 + 10 |
| L* | Control | 62.42 ± 2.16 | 60.83 ± 1.83 | 65.33 ± 1.99 a | 66.08 ± 0.38 a | 69.39 ± 0.65 a | 69.86 ± 0.32 a | 70.06 ± 0.19 a | 67.28 ± 0.41 ab |
| | 0.5 ppm | 62.42 ± 2.16 | 52.08 ± 1.83 | 60.58 ± 2.73 ab | 62.00 ± 1.19 ab | 71.28 ± 0.93 a | 65.50 ± 0.74 ab | 62.42 ± 0.48 b | 66.67 ± 0.78 b |
| | 1.0 ppm | 62.42 ± 2.16 | 51.25 ± 2.77 | 50.42 ± 1.77 b | 62.47 ± 1.13 ab | 70.22 ± 0.83 a | 64.75 ± 0.77 ab | 59.86 ± 0.92 b | 70.94 ± 0.13 a |
| | 2.0 ppm | 62.42 ± 2.16 | 49.83 ± 2.91 | 52.17 ± 1.29 b | 58.00 ± 1.23 b | 62.00 ± 1.65 b | 62.75 ± 0.31 bc | 61.92 ± 0.76 b | 65.03 ± 1.14 b |
| | 4.0 ppm | 62.42 ± 2.1 a | 54.58 ± 2.26 | 52.50 ± 0.66 b | 58.06 ± 1.00 b | 57.08 ± 0.96 b | 57.92 ± 1.95 c | 59.81 ± 0.61 b | 63.86 ± 0.68 b |
| a* | Control | −16.17 ± 0.77 | −7.67 ± 2.00 a | −6.00 ± 4.22 | 9.94 ± 1.05 a | 3.47 ± 2.02 a | 8.75 ± 0.31 a | 14.69 ± 1.41 a | 22.28 ± 0.62 a |
| | 0.5 ppm | −16.17 ± 0.77 | −22.83 ± 2.19 b | −11.42 ± 1.58 | −4.53 ± 1.54 b | 4.39 ± 0.85 a | 5.94 ± 1.41 ab | 13.00 ± 2.58 a | 24.44 ± 0.08 a |
| | 1.0 ppm | −16.17 ± 0.77 | −23.67 ± 1.20 b | −17.92 ± 0.18 | −10.03 ± 1.08 bc | −5.19 ± 0.23 b | 3.19 ± 0.82 ab | 7.28 ± 0.33 ab | 17.19 ± 0.84 b |
| | 2.0 ppm | −16.17 ± 0.77 | −20.08 ± 2.79 ab | −17.25 ± 0.82 | −12.89 ± 0.65 c | −5.81 ± 2.12 b | 1.81 ± 1.29 b | 12.89 ± 1.56 a | 15.81 ± 0.75 b |
| | 4.0 ppm | −16.17 ± 0.77 | −15.75 ± 2.60 ab | −17.75 ± 1.63 | −11.50 ± 1.93 bc | −11.11 ± 0.92 b | −5.03 ± 0.94 c | 1.53 ± 1.19 b | 4.61 ± 0.83 c |
| b* | Control | 66.33 ± 1.09 | 75.75 ± 1.36 a | 73.67 ± 2.98 a | 74.36 ± 1.11 a | 76.06 ± 1.07 a | 76.53 ± 0.49 | 70.06 ± 0.72 ab | 78.22 ± 0.35 a |
| | 0.5 ppm | 66.33 ± 1.09 | 62.33 ± 1.77 ab | 56.58 ± 3.13 b | 66.69 ± 2.83 ab | 72.94 ± 1.36 ab | 73.78 ± 3.29 | 69.89 ± 0.99 ab | 79.11 ± 0.53 a |
| | 1.0 ppm | 66.33 ± 1.09 | 60.33 ± 1.88 b | 49.83 ± 4.72 b | 62.89 ± 1.14 b | 70.58 ± 0.38 ab | 75.17 ± 1.19 | 64.14 ± 1.16 c | 76.53 ± 0.47 ab |
| | 2.0 ppm | 66.33 ± 1.09 | 62.75 ± 3.89 ab | 47.83 ± 0.92 b | 58.50 ± 0.31 b | 64.78 ± 3.19 bc | 74.00 ± 1.12 | 71.28 ± 1.32 a | 70.75 ± 2.36 bc |
| | 4.0 ppm | 66.33 ± 1.09 | 69.08 ± 3.18 ab | 58.42 ± 1.41 ab | 60.44 ± 1.76 b | 55.17 ± 1.91 c | 70.17 ± 2.45 | 65.31 ± 0.66 bc | 68.03 ± 1.01 c |
| Firmness (N) | Control | 55.49 ± 10.57 | 24.86 ± 6.65 b | 27.48 ± 4.74 | 11.51 ± 1.72 c | 4.52 ± 0.39 b | 7.81 ± 1.83 bc | 3.96 ± 1.11 b | 1.97 ± 0.26 c |
| | 0.5 ppm | 55.49 ± 10.57 | 36.61 ± 2.54 ab | 43.66 ± 10.85 | 11.85 ±2.91 c | 11.61 ± 2.49 ab | 4.38 ±0.35 c | 8.40 ±2.03 ab | 5.56 ± 2.70 b |
| | 1.0 ppm | 55.49 ± 10.57 | 74.86 ± 6.09 a | 67.14 ± 12.23 | 22.12 ± 1.03 bc | 39.29 ± 9.86 a | 9.97 ± 3.13 abc | 15.83 ± 3.67 ab | 8.76 ± 1.13 b |
| | 2.0 ppm | 55.49 ± 10.57 | 33.91 ± 14.59 ab | 56.48 ± 12.75 | 29.97 ± 1.56 b | 26.99 ± 5.35 ab | 40.97 ± 11.25 a | 33.98 ± 8.98 a | 20.48 ± 7.29 a |
| | 4.0 ppm | 55.49 ± 10.57 | 36.61 ± 2.54 ab | 51.97 ± 8.81 | 46.69 ± 3.22 a | 39.37 ±3.11 a | 40.26 ±5.25 ab | 22.73 ±5.58 ab | 14.99 ±3.07 a |
| SSC (%) | Control | 15.17 ± 0.26 | 14.83 ± 0.29 | 15.07 ± 0.09 | 14.90 ± 0.22 | 14.37 ± 0.22 ab | 13.00 ± 0.27 a | 14.25 ± 0.37 | 14.10 ± 0.10 |
| | 0.5 ppm | 15.17 ± 0.26 | 14.05 ± 0.10 | 15.55 ± 0.36 | 13.85 ± 0.38 | 13.78 ± 0.06 ab | 11.80 ± 0.31 b | 12.87 ± 0.28 | 13.82 ± 0.22 |
| | 1.0 ppm | 15.17 ± 0.26 | 14.53 ± 0.09 | 15.30 ± 0.18 | 14.23 ± 0.18 | 13.50 ± 0.19 b | 12.58 ± 0.13 ab | 13.17 ± 0.47 | 12.78 ± 0.42 |
| | 2.0 ppm | 15.17 ± 0.26 | 15.67 ± 0.33 | 15.75 ± 0.70 | 14.88 ± 0.70 | 14.92 ± 0.35 ab | 12.22 ± 0.09 ab | 13.75 ± 0.18 | 13.90 ± 0.47 |
| | 4.0 ppm | 15.17 ± 0.26 | 14.87 ± 0.62 | 14.32 ± 0.58 | 14.32 ± 0.58 | 15.18 ± 0.31 a | 12.82 ± 0.08 ab | 13.73 ± 0.18 | 13.35 ± 0.40 |
| Titratable acidity (%) | Control | 0.31 ± 0.06 | 0.29 ± 0.01 | 0.28 ± 0.00 | 0.32 ± 0.03 | 0.34 ± 0.01 | 0.27 ± 0.01 b | 0.27 ± 0.01 b | 0.31 ± 0.01 ab |
| | 0.5 ppm | 0.31 ± 0.06 | 0.32 ± 0.01 | 0.33 ± 0.01 | 0.30 ± 0.03 | 0.29 ± 0.03 | 0.40 ± 0.06 a | 0.29 ± 0.02 b | 0.26 ± 0.01 b |
| | 1.0 ppm | 0.31 ± 0.06 | 0.31 ± 0.02 | 0.29 ± 0.00 | 0.24 ± 0.01 | 0.36 ± 0.01 | 0.38 ± 0.04 a | 0.40 ± 0.02 a | 0.26 ± 0.01 b |
| | 2.0 ppm | 0.31 ± 0.06 | 0.25 ± 0.01 | 0.28 ± 0.04 | 0.26 ± 0.01 | 0.33 ± 0.05 | 0.29 ± 0.04 ab | 0.36 ± 0.02 ab | 0.27 ± 0.00 b |
| | 4.0 ppm | 0.31 ± 0.06 | 0.31 ± 0.03 | 0.26 ± 0.02 | 0.24 ± 0.02 | 0.29 ± 0.00 | 0.32 ± 0.01 ab | 0.32 ± 0.03 ab | 0.34 ± 0.02 a |

* Papaya fruits were treated by 0 (control), 0.5, 1.0, 2.0, and 4.0 ppm 1-MCP, and the fruits were stored at 10 ± 0.5 °C and relative humidity (RH) 85% ± 2% for 14 days, and then transferred to 20 ± 0.5 °C and RH 85% ± 2% for 10 days. The values are expressed as the mean ± SEM ($n$ = 3), and the tests carried out three independent times. Values in the same column followed by different letters were significantly different at $p < 0.05$ at the same timepoint.

### 3.2. Weight Loss

Papayas of all groups lost weight continuously during cold storage and at room temperature. The control level was significantly higher than all 1-MCP-treated groups but there were no significant differences among 1-MCP-treated groups (Figure 3). The weight loss of the control group was the highest, approximately two times higher than the 1.0 ppm group, and was lowest on days 14, 14 + 2, and 14 + 4. The weight loss of 2.0 and 4.0 ppm 1-MCP trended higher than that of 0.5 and 1.0 ppm, but not significantly. The 1.0 ppm treatment during cold storage and the 0.5 ppm treatment at room temperature showed the slowest weight loss using linear regression model (Table 2).

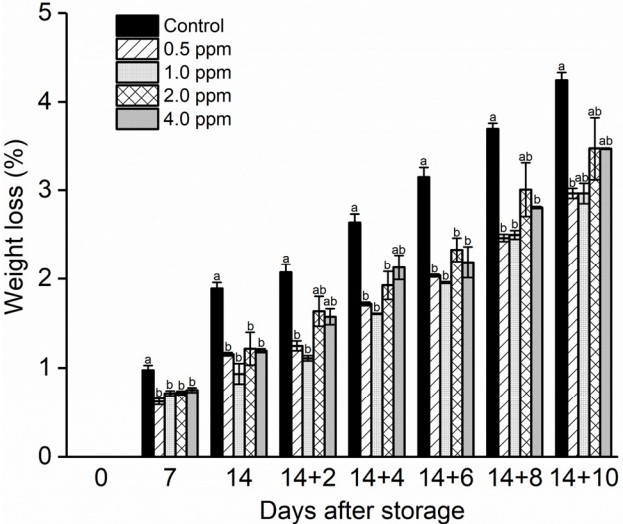

**Figure 3.** Effect of 1-MCP treatment on papaya weight loss during cold storage and shelf life. Papaya fruits were treated by 0 (control), 0.5, 1.0, 2.0, and 4.0 ppm 1-MCP, and the fruits were stored at $10 \pm 0.5\,^{\circ}$C and relative humidity (RH) $85\% \pm 2\%$ for 14 days, and then transferred to $20 \pm 0.5\,^{\circ}$C and RH $85\% \pm 2\%$ for 10 days. Each value is the mean of three replicates. The vertical bars represent the standard errors of the means. The different letters indicate significant differences (ANOVA, $p < 0.05$) between different groups at the same timepoint, according to Tukey's comparisons test.

**Table 2.** Linear regressions of papaya weight loss during cold storage and shelf life *.

| Group | Cold Storage | | Shelf Life | |
|---|---|---|---|---|
| Control | y = 0.135x + 0.0102 | $R^2 = 0.9997$ | y = 0.541x + 1.5358 | $R^2 = 0.9998$ |
| 0.5 ppm | y = 0.0825x + 0.0145 | $R^2 = 0.9981$ | y = 0.4175x + 0.8318 | $R^2 = 0.9952$ |
| 1.0 ppm | y = 0.0666x + 0.083 | $R^2 = 0.9132$ | y = 0.4595x + 0.6469 | $R^2 = 0.9968$ |
| 2.0 ppm | y = 0.0869x + 0.037 | $R^2 = 0.9890$ | y = 0.475x + 1.0472 | $R^2 = 0.9807$ |
| 4.0 ppm | y = 0.0852x + 0.0512 | $R^2 = 0.9783$ | y = 0.4464x + 1.092 | $R^2 = 0.9488$ |

* Papaya fruits were treated by 0 (control), 0.5, 1.0, 2.0, and 4.0 ppm 1-MCP, and the fruits were stored at $10 \pm 0.5\,^{\circ}$C and relative humidity (RH) $85\% \pm 2\%$ for 14 days, and then transferred to $20 \pm 0.5\,^{\circ}$C and RH $85\% \pm 2\%$ for 10 days. The tests carried out three independent times. Y represents weight loss, X refers to days after cold storage or shelf life, and the slope shows the daily weight loss of each group.

Weight loss is a major determinant of storage quality of papaya fruit [23]. Fruit respiration takes place through the pericarp and the stomata to achieve atmosphere exchange. Therefore, weight loss of fruit is affected by its physiological status. Various studies have shown that 1-MCP can be effective in retarding papaya weight loss [24,25]. The present result was consistent with previous findings, although the reduction in fruit weight loss was not related to the concentration of 1-MCP.

### 3.3. Firmness

1-MCP significantly delayed fruit softening during cold storage and, in a dose-dependent manner, resulted in firmer fruit during storage (Table 1). The firmness of the control fruit decreased by 50% at day 14 compared to day 0 and reached optimum firmness for eating quality (Table 3) after 2 days at room temperature. In contrast, the 4.0 ppm 1-MCP-treated fruit showed the highest firmness after cold storage, which was 1.89 times higher than the control fruit. During storage at room temperature, firmness in all groups decreased significantly. These results demonstrate that 1-MCP can effectively mitigate firmness loss.

**Table 3.** Days fruit reached edible softness and lost marketability.

| Dose of 1-MCP (ppm) | Edible Stage | Unmarketable * |
|---|---|---|
| 0 (Control) | 7 | 14 + 2 |
| 0.5 | 14 + 2 | 14 + 4 |
| 1.0 | 14 + 2 | 14 + 6 |
| 2.0 | 14 + 4 | >14 + 10 |
| 4.0 | 14 + 8 | >14 + 10 |

* The edible stage and unmarketable date were evaluated as the pulp firmness reached 22 N and 10 N respectively, according to a previous study [26].

Firmness is the attribute most closely associated with tolerance to compression, bruise, and microbial contamination. Papaya reaches edible softness when the firmness reaches about 22 N or below [26]. Fruit becomes fragile and unmarketable when flesh firmness reaches 10 N or lower. In our study, changes in firmness were related to 1-MCP dosage. Treatment with 1-MCP significantly extended papaya marketability (Table 3). Fruit softening is the result of changes in cell wall composition in response to cell-wall-modifying enzymes [27]. Typical of climacteric fruit, papaya produces a large amount of ethylene during storage, which promotes fruit softening [8]. As an ethylene receptor competitive inhibitor, 1-MCP irreversibly competes with the ethylene receptor of the fruit, reducing ethylene's normal phytohormonal effects. In addition, 1-MCP is known to regulate fruit ethylene metabolism and reduce the rate of ethylene production [28]. This study demonstrated that 1-MCP was effective at room temperature. Fruits responded differently to different 1-MCP concentrations. When 1-MCP concentration is too low, it does not improve postharvest quality, whereas, when concentration is too high, physiological disorders including peel browning, ripening disorder, and off-flavors can occur [14].

Previous studies on 1-MCP treatment used 1.0 ppm or lower doses which substantially extended the storage life of papayas and other fruits [8,29,30]. The different responses to 1-MCP doses were mainly due to fruit ripening stages at the time of application; papayas harvested at mature green or breaker stages [9] or pears before ripening initialization [10] respond to low 1-MCP efficiently, perhaps "permanently" blocking ripening. On the other hand, after 1-MCP application, papayas harvested at the half-yellow stage [9] or partially ripened pears [11] continued to ripen, but did so at a reduced rate. Zhang et al. indicated that internal ethylene levels may contribute to the divergent sensitivities of some climacteric fruits to 1-MCP applied after initiation of ripening [31].

### 3.4. Soluble Solid Content and Titratable Acidity

Total soluble solid content (SSC) decreased continuously in all groups over the course of the experiment (Table 1). 1-MCP had no significant impact on papaya SSC when compared to the control, either during cold storage or at room temperature. On day 7, the SSC in 2.0 ppm 1-MCP group was the highest at 15.66% (Table 1). The titratable acidity (TA) was relatively stable during storage (Table 1). At the final timepoint, the TA of the 4.0 ppm group was the highest, which was 13.33% higher than that of the control; significant differences were observed after 6 days of shelf life.

The reduction in SSC during storage is related to the increase in respiration and sugar consumption for energy [32]. Previous studies showed that 1-MCP can maintain the SSC and organic acid content of papaya by inhibiting ethylene production [25]. However, our results showed no significant difference in SSC content in the 1-MCP-treated groups in the cold storage and the beginning of the shelf-life, which is similar to previous findings in 'Eksotika' papaya by Ding et al. [33]. This may be due to the relatively riper maturity stage (half-yellow) at harvest, which may have diminished the effects of 1-MCP on respiration.

*3.5. Decay Rate*

No decay occurred in any group during the cold storage, but decay started to appear after 2 days at room temperature. However, there was no significant difference between the groups for decay incidence during the entire shelf-life period (Figure 4).

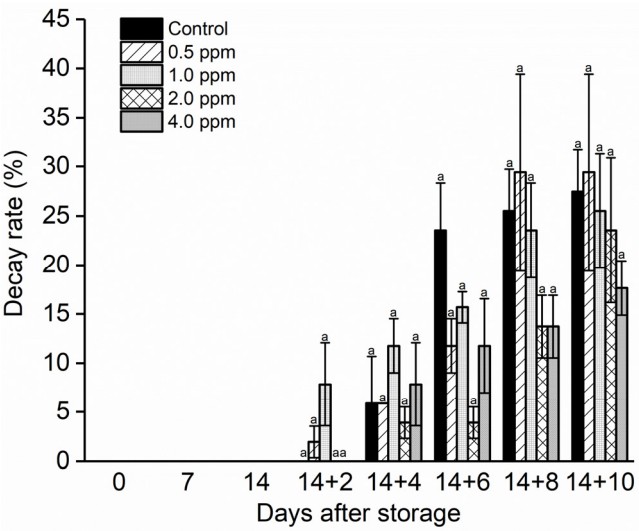

**Figure 4.** Effect of 1-MCP treatment on papaya decay rate during storage and shelf life. Papaya fruits were treated by 0 (control), 0.5, 1.0, 2.0, and 4.0 ppm 1-MCP, and the fruits were stored at $10 \pm 0.5$ °C and relative humidity (RH) 85% $\pm$ 2% for 14 days, and then transferred to $20 \pm 0.5$ °C and RH 85% $\pm$ 2% for 10 days. Each value is the mean of three replicates. The vertical bars represent the standard errors of the means. The letters indicate significant analysis result (ANOVA, $p < 0.05$) between different groups at the same timepoint, according to Tukey's comparisons test.

Papaya is susceptible to a series of postharvest pathogenic microorganisms during storage, such as *Colletotrichum gloeosporioides* and *Fusarium* sp. [34], which also increases potential food safety problems. 1-MCP itself has direct inhibitory effect on pathogens [35], and less mature fruit are more resistant to pathogens due to the presence of antimicrobial substances such as phenolic acids [36]. Previous studies showed that the decay incidence of papaya decreased with increasing 1-MCP concentration [25]; however, the direct antifungal activity of 1-MCP is limited. Tropical fruit are sensitive to pathogen invasion, and infected lesions develop quickly. This study only measured decay rate in order to illustrate the decrease in decay severity; however, further study is needed to measure the degree of disease in order to characterize the disease situation more completely.

Ethylene is an important hormone for the regulation of plant maturation and senescence, and it is also involved in the plant's disease resistance response, acting as a signal molecular participant in defense responses [37]. When fruit is infected by the pathogen, its ethylene biosynthesis increases significantly, and the receptors respond to the ethylene and further activate downstream responses [38]. As an ethylene antagonist, 1-MCP inhibits the response to ethylene, which may affect the overall response to disease [39]. The controlled-release 1-MCP used in this study inhibited the decay, but there was no significant difference among the groups, which may be related to the fruit variety, the sensitivity to

1-MCP/ethylene, the release rate of 1-MCP, storage conditions, etc. Further research is needed to confirm this.

### 3.6. Sensory Evaluation

As shown in Figure 5, the sweetness of all the groups was not significantly different. There was no significant difference in sourness among the control, 1.0 ppm, and 2.0 ppm 1-MCP@α-cyclodextrin clathrates, but the sourness of the 4.0 ppm group was the highest at 1.54 times the score of the control group. The papaya flavor score of the 1.0 ppm treatment group was the highest, but the differences between the treatments were not significant. The control group had the highest score for off-flavors, followed by the 0.5 ppm group. It is noteworthy that the score of off-flavor in the 1.0 ppm treatment group was 0, while, in the 2.0–4.0 ppm 1-MCP groups, it was slightly higher. The flesh hardness of the 2.0 ppm and 4.0 ppm 1-MCP groups was the highest, whereas the control group was the lowest, which was consistent with the fruit penetrometer results. The overall score of the 4.0 ppm 1-MCP group was the highest, followed by the 1.0 ppm 1-MCP group.

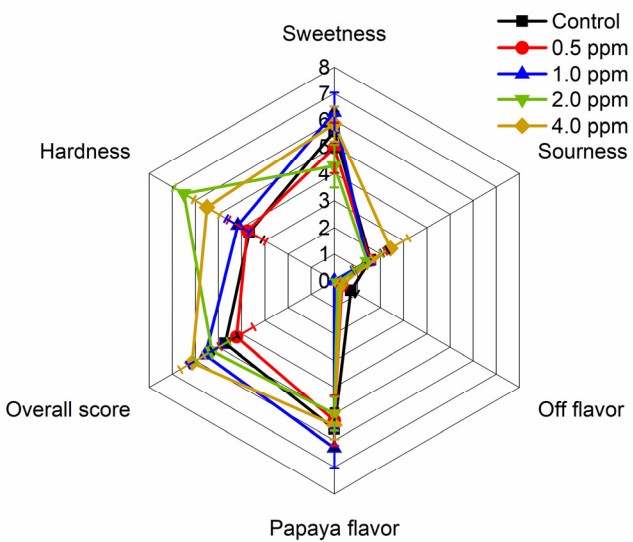

**Figure 5.** Effect of 1-MCP treatment on papaya sensory evaluation after the shelf life. Sensory evaluations were taken on papaya fruit after treating with 0 (control), 0.5, 1.0, 2.0, and 4.0 ppm 1-MCP, stored at 10 ± 0.5 °C and relative humidity (RH) 85% ± 2% for 14 days and held at 20 ± 0.5 and RH 85% ± 2% for 10 days. Each value is the mean of seven replicates. The vertical bars represent the standard errors of the means.

It is common that, for many climacteric fruits, the flavor and overall sensory quality reach their peak after ripening, and overripe fruits, such as bananas, develop off-flavor due to an accumulation of ethanol and ethyl esters [40]. The 1-MCP@α-cyclodextrin clathrates in this study slowed the ripening and senescence in papayas and, thus, imparted a lower off-flavor. The initial maturity of the papayas used in this experiment was half-yellow. Use of 1-MCP with maturity less than quarter-yellow can affect overall quality and normal ripening [25] or result in other physiological disorders such as unfavorable fruit flavor [14].

The sensory quality of fruit is determined by a series of factors such as appearance, taste, and aroma [41,42]. For taste, sweetness is the most obvious indicator of papaya ripeness, and it is mainly determined by the sugar content in the fruit [43]. In this experiment, the sensory sweetness was positively correlated with SSC content (Table 4). However, the difference in sweetness between groups was not significant, and this might have been due to all fruit being normally mature at the end of the experiment, resulting high sugar content. No unripening disorders were observed. There are a variety of organic acids in fruits, which can be a source of acidic taste, but which also play an important role in the respiratory metabolism of the fruit [44]. The main organic acid in papaya is citric acid,

which would continue to decline in concentration after harvest. The higher TA content indicates that 1-MCP@α-cyclodextrin clathrate inhibited the consumption rate of organic acids. The high positive correlation coefficients between TA content and sourness also demonstrated this (Table 4). In addition to sugar and acid, there are also volatile substances such as aldehydes and ketones interacting with other components to affect the flavor of papaya during ripening [45]. In our study, the papaya flavor score at 1.0–2.0 ppm was found to be the highest. This might be due to the fact that the fruit in the 0 to 0.5 ppm 1-MCP groups were overripe at the time of sensory evaluation (14 + 10 days); on the other hand, an overdose of 1-MCP, such as 4 ppm, could partially retard the ripening process and flavor development [10,30]. However, the sensory quality of papaya is complicated, and further details need to be explored in the future.

**Table 4.** Pearson correlation coefficients between firmness, SSC, TA and sensory parameters.

|  | Firmness | SSC | TA | Sweetness | Sourness | Off-Flavor | Papaya Flavor | Hardness | Overall Score |
|---|---|---|---|---|---|---|---|---|---|
| Firmness | 1 | | | | | | | | |
| SSC | −0.168 | 1 | | | | | | | |
| TA | 0.238 | 0.106 | 1 | | | | | | |
| Sweetness | −0.331 | 0.714 | 0.309 | 1 | | | | | |
| Sourness | 0.399 | −0.160 | 0.865 | 0.335 | 1 | | | | |
| Off-flavor | −0.554 | 0.731 | 0.464 | −0.044 | 0.123 | 1 | | | |
| Papaya flavor | −0.405 | −0.773 | −0.118 | 0.883 | −0.118 | −0.255 | 1 | | |
| Hardness | 0.968 | 0.022 | 0.198 | −0.485 | 0.253 | −0.441 | −0.509 | 1 | |
| Overall score | 0.691 | 0.615 | 0.542 | 0.431 | 0.623 | −0.454 | 0.268 | 0.578 | 1 |

This research reported the edible quality in response to humidity-triggered con-trolled-release 1-MCP clathrate; further research needs to explore the physiological (ethylene and respiration pathway, cell membrane indicators, etc.) changes amd chemical content (such as carotenoids, ascorbic acid, total phenols, total flavonoids, etc.) changes, to compare with traditional 1-MCP/other commercial 1-MCP treatments. Its large-scale application in the industry still needs stronger theoretical basis.

## 4. Conclusions

The humidity-triggered controlled-release 1-MCP delayed the ripening and senescence processes in papayas, including slowing softening, reducing weight loss, peel de-greening, and decay, and improving sensory quality, with no observed negative side effects. Among the treatments, 1.0–2.0 ppm treatment is the recommended level to optimize storage life. HarvestHold® allows papayas to ripen more slowly and does not significantly impact the sensory attributes associated with papayas. The controlled-release 1-MCP product is an effective and convenient delivery system for extending shelf life of 'Rainbow' papaya for commercial application, transportation, retailers, and the consumer. The postharvest physiology and nutrient content changes in response to controlled-release 1-MCP need to be further studied.

**Author Contributions:** Conceptualization, X.S. and N.S.; methodology, X.S. and C.S.; software, C.S.; validation, C.S. and X.S.; formal analysis, C.S. and X.S.; investigation, C.S. and X.S.; resources, X.S. and N.S.; data curation, C.S., N.S., and X.S.; writing—original draft preparation, C.S., X.S., and J.B.; writing—review and editing, C.S., M.M.W., P.A.F., N.S., J.B., and X.S.; visualization, C.S.; supervision, X.S.; project administration, X.S., M.M.W., P.A.F., and N.S.; funding acquisition, X.S. All authors have read and agreed to the published version of the manuscript.

**Funding:** This research received no external funding.

**Data Availability Statement:** The data used to support the findings of this study can be made available by the corresponding author upon request.

**Acknowledgments:** The authors thank Xiaohua Wu, Keegan Duff, and Melissa Postler for their technical support during the experiments. This research was supported in part by an appointment to the Agricultural Research Service (ARS) Research Participation Program administered by the Oak Ridge Institute for Science and Education (ORISE) through an interagency agreement between the US Department of Energy (DOE) and the US Department of Agriculture (USDA). ORISE is managed by ORAU under DOE contract number DE-SC0014664. All opinions expressed in this paper are the authors' and do not necessarily reflect the policies and views of the USDA, DOE, or ORAU/ORISE. USDA is an equal-opportunity provider and employer.

**Conflicts of Interest:** The authors declare no conflict of interest.

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
