# Peer review of "Effect of Humidity-Triggered Controlled-Release 1-Methylcyclopropene (1-MCP) on Postharvest Quality of Papaya Fruit"

_horticulturae, doi:10.3390/horticulturae9101062_

Round 1

Reviewer 1 Report

In the manuscript, authors investigated the effect 1-MCP treatment on maintaining postharvest quality of papaya fruit. Although the function of 1-MCP on maintenance postharvest quality of fruit is clear, the 1-MCP product used in this experiment is clathrates of 1-MCP with α-cyclodextrin, which is different from the traditional one. Therefore, it is interesting to discover the function of a new 1-MCP product. However, there are a few questions need to be solved before the further consideration.

1.     Line 126-128, this section should be "decay rate". The authors should provide more details about how to measure the decay, it was based on the lesion diameter? How big of the lesion was defined as "decayed"? Will it be better to combine the lesion size and number to calculate the decay index?

2.     Line 132, what is "firmness" mean in this section? It should be revised to "hardness".

3.     Line 219, the author should provied the meaning of linear model slope

4.     Line 283, "storage" should be revised to "shelf-life".

5.     Line 294, the author should provide more details about the role of ethylene involved in fruit resistance, and how 1-MCP may block this resistance-related pathway.

6.     In section 3.6, the author should analyze the correlation between firmness, SSC, TA, and sensory quality. Why use the fruit from the last day of shelf life? Based on Table 3, the fruit from the control was not in the appropriate edible stage.

Author Response

Comment 1: In the manuscript, authors investigated the effect 1-MCP treatment on maintaining postharvest quality of papaya fruit. Although the function of 1-MCP on maintenance postharvest quality of fruit is clear, the 1-MCP product used in this experiment is clathrates of 1-MCP with α-cyclodextrin, which is different from the traditional one. Therefore, it is interesting to discover the function of a new 1-MCP product. However, there are a few questions need to be solved before the further consideration.

Response 1: Thank you very much for your positive comments. We have revised the manuscript according to your suggestions. Our point-to-point responses are as follows.

Comment 2: Line 126-128, this section should be "decay rate". The authors should provide more details about how to measure the decay, it was based on the lesion diameter? How big of the lesion was defined as "decayed"? Will it be better to combine the lesion size and number to calculate the decay index?

Response 2: We have revised the “Decay incidence” to “Decay rate”. The evaluation of fruit decay was based on the visible lesion, since the fruit with visible lesions had already lost their commercial values, we just defined them as “decayed”. We agree with the decay index will be better to reflex the decay severity, but we did not design it in this research. In the discussion, we indicated that the decay index would be measured in the further study (Lines 304-307).

Comment 3: Line 132, what is "firmness" mean in this section? It should be revised to "hardness".

Response 3: We have revised “firmness” to “hardness” in this section, and all the others in the sensory section.

Comment 4: Line 219, the author should provied the meaning of linear model slope

Response 4: We have provided the meaning of linear model slops in the table’s caption.

Comment 5: Line 283, "storage" should be revised to "shelf-life".

Response 5: We have revised to “shelf-life” in the manuscript.

Comment 6: Line 294, the author should provide more details about the role of ethylene involved in fruit resistance, and how 1-MCP may block this resistance-related pathway.

Response 6: We have provided some discussion regarding how ethylene pathway involved in fruit resistance against pathogen (Lines 308-317).

Comment 7: In section 3.6, the author should analyze the correlation between firmness, SSC, TA, and sensory quality. Why use the fruit from the last day of shelf life? Based on Table 3, the fruit from the control was not in the appropriate edible stage.

Response 7: We analyzed the correlation between firmness, SSC, TA, and sensory quality in Table 4. The sensory study was conducted to simulate the fruit after transit and shelf-life at room temperature for 10 days. So, this section was to demonstrate even after the whole cold storage and shelf-life, the treated fruit was still in the edible stage. Although the fruit from the control was not in the appropriate edible stage at day 14+10, but it demonstrated that 1-MCP prolonged the fruit shelf life.

Reviewer 2 Report

The manuscript entitled “Effect of humidity-triggered controlled-release 1-methylcyclopropene (1-MCP) on postharvest quality of papaya fruit” centered on the use of humidity-triggered controlled-release 1-methylcyclopropene (1-MCP) sheets to enhance the quality of 'Rainbow' papayas. The experiment involved a cold storage phase followed by shelf simulation, and different sizes of 1-MCP sheets were utilized to achieve varying concentrations. The findings demonstrate that 1-MCP treatment effectively preserves fruit quality by delaying softening, reducing weight loss, and preventing peel color deterioration, all while maintaining the fruit's physiological health. Notably, papayas treated with 1.0-2.0 ppm 1-MCP received the highest scores for desirable flavor and sweetness and the lowest scores for undesirable off-flavor. The study concludes that humidity-triggered controlled-release 1-MCP sheets offer a practical and efficient solution for regulating papaya ripening during postharvest, ultimately resulting in economic benefits for producers and consumers alike.

The work is interesting and has an economic impact on fruit transportation and marketing. However, some points should be addressed to enhance the manuscript's clarity for the reader.

There is a vast body of literature on the use of 1-MCP on papaya fruit. Why did you not include some of them in your introduction? Please expand the references for the section on 1-MCP's effects on papaya fruit in the introduction to highlight the importance of your new technology in comparison with previous approaches.

In line 39: use “receptors” instead of “action”

Why fruit ethylene was not measured in this trial since it is the main factor of the1-MCP treatment effect.

You used different 1-MCP concentrations, but it is not clear how the fruit received the exact dose of the 1-MCP. Did you store fruit in different sealed chambers? The 1-MCP gas could not be held in the same container, and it will be all around the place if you put them in one room. More clarification is needed.

In table 1: I suggest delete the letters in any category doe not show significant effect.

Author Response

Comment 1: The manuscript entitled “Effect of humidity-triggered controlled-release 1-methylcyclopropene (1-MCP) on postharvest quality of papaya fruit” centered on the use of humidity-triggered controlled-release 1-methylcyclopropene (1-MCP) sheets to enhance the quality of 'Rainbow' papayas. The experiment involved a cold storage phase followed by shelf simulation, and different sizes of 1-MCP sheets were utilized to achieve varying concentrations. The findings demonstrate that 1-MCP treatment effectively preserves fruit quality by delaying softening, reducing weight loss, and preventing peel color deterioration, all while maintaining the fruit's physiological health. Notably, papayas treated with 1.0-2.0 ppm 1-MCP received the highest scores for desirable flavor and sweetness and the lowest scores for undesirable off-flavor. The study concludes that humidity-triggered controlled-release 1-MCP sheets offer a practical and efficient solution for regulating papaya ripening during postharvest, ultimately resulting in economic benefits for producers and consumers alike. The work is interesting and has an economic impact on fruit transportation and marketing. However, some points should be addressed to enhance the manuscript's clarity for the reader.

Response 1: Thank you for the positive comments. We have revised the manuscript according to your suggestions. Our point-to-point responses are as follows.

Comment 2: There is a vast body of literature on the use of 1-MCP on papaya fruit. Why did you not include some of them in your introduction? Please expand the references for the section on 1-MCP's effects on papaya fruit in the introduction to highlight the importance of your new technology in comparison with previous approaches.

Response 2: We have included some previous research regarding the application of 1-MCP on papaya fruit, and the importance of our new technology. (Lines 55-58)

Comment 3: In line 39: use “receptors” instead of “action”

Response 3: We have revised this word in the manuscript.

Comment 4: Why fruit ethylene was not measured in this trial since it is the main factor of the1-MCP treatment effect.

Response 4: We measured the fruit internal atmosphere change in the pre-experiment, which contains ethylene, CO2, and O2. However, we didn’t notice significant differences among the groups. This might be because the 1-MCP molecular cannot penetrate the pulp to the core, so the internal ethylene level is not significantly affected. Previous study observed increased internal ethylene level in response to 1-MCP treatment (Façanha et al., 2019). This may due to the auto-inhibitory ethylene action decreased, in the presence of 1-MCP bound to its receptor leading to higher ACC levels and increased ethylene biosynthesis (Philosoph-Hadas et al., 1994). Similar symptom also observed in 1-MCP-treated grapefruit (McCollum & Maul, 2007) and coriander (Jiang et al., 2002). We have supplied some discussion regarding this, and plan to measure the ethylene release rate in further study.

Reference:

Façanha, R. V., Spricigo, P. C., Purgatto, E., & Jacomino, A. P. (2019). Combined application of ethylene and 1-methylcyclopropene on ripening and volatile compound production of 'Golden' papaya. Postharvest Biology and Technology, 151, 160-169. https://doi.org/10.1016/j.postharvbio.2019.02.005

Jiang, W., Sheng, Q., Zhou, X., Zhang, M., & Liu, X. (2002). Regulation of detached coriander leaf senescence by 1-methylcyclopropene and ethylene. Postharvest Biology and Technology, 26(3), 339-345. https://doi.org/10.1016/S0925-5214(02)00068-6

McCollum, G., & Maul, P. (2007). 1-Methylcyclopropene inhibits degreening but stimulates respiration and ethylene biosynthesis in grapefruit. HortScience, 42(1), 120-124. https://doi.org/10.21273/HORTSCI.42.1.120

Philosoph-Hadas, S., Meir, S., & Aharoni, N. (1994). Role of ethylene in senescence of watercress leaves. Physiologia Plantarum, 90(3), 553-559. https://doi.org/10.1111/j.1399-3054.1994.tb08814.x

Comment 5: You used different 1-MCP concentrations, but it is not clear how the fruit received the exact dose of the 1-MCP. Did you store fruit in different sealed chambers? The 1-MCP gas could not be held in the same container, and it will be all around the place if you put them in one room. More clarification is needed.

Response 5: The 1-MCP sheet was cut into different sizes according to the manufacturer's guidance, and the concentration was calculated based on the product’s total 1-MCP amount in the sheet and the release rate. The 1-MCP concentrations are positively linear correlated with the sheet sizes under the constant temperature and relative humidity. The fruit were stored separately with one treatment per room.

Reference:

Wood, W; Sarageno, J.F; Keute, J; Lundgren A. Compositions and methods for differential release of 1-methylcyclopropene. US Patent 11492419B2, filed 17 February 2021, issued 8 November 2022.

Comment 6: In table 1: I suggest delete the letters in any category doe not show significant effect.

Response 6: We have deleted the significance labels of the categories that do not show significant differences in Table 1.

Reviewer 3 Report

In this study, authors investigated the effect of humidity-triggered controlled-release 1-MCP on postharvest quality of papaya fruit. It is interesting to explore the impact of different size of 1-MCP sheet on the cold stage and shelf life of papaya. However, there are still some questions need to be answered. 

1.      In the abstract, the authors said the 1.0-2.0 ppm 1-MCP treated fruit received the highest score for papaya flavor and sweetness (L.21), but from the Figure 5, we could see that 2.0 ppm 1-MCP treated fruit had the lowest score for sweetness, not the highest. Meantime, this study didn’t conclude which ppm 1-MCP treated papaya has the highest quality in after the cold stage and shelf life.

2.      No matter in cold stage or shelf life, the humidity was the same. While the temperature was different, from 10 °C to 20 °C. I couldn’t understand what the humidity-triggered is. Because no change occurred in humidity.

3.      Regarding the linear regressions representing the daily weight loss of the fruit (Table 2), in the cold stage, there were only two sample date. Could two points be used to plot a linear regression curve?

4.      In the conclusions, the authors recommend 1.0-2.0 ppm treatment could optimize storage life of papaya, but from the Figure 4, we could find that under 1.0 ppm treatment, the decay rate of papaya reached nearly 10%, which was the highest decay rate. In this view, I don’t think 1 ppm could be recommended to optimize storage life of papaya.

5.      Authors should consider these indexes comprehensively and make a more scientific conclusion. 

Minor issues are following:

1.      In table 1, there appeared storage time like 16+8, 16+10? On day 14+10, the firmness of Papaya samples is significantly different, but the values by different letters represented significantly difference were the same. The same issue occurred in acidity index. It’s impossible that no difference among different groups.

2.      L208 Table 1 should be changed to Table 2.

3.      Table 2 and 3 have the same title.

4.      References should be formatted.

Reviewer 4 Report

The manuscript evaluated the effects of humidity-triggered controlled-release 1-methylcyclo-2 propene (1-MCP) clathrates with α-cyclodextrin on postharvest quality of papaya fruit. The manuscript presents an interesting read, abstract is properly written and appropriately reflects the results of the study. The introduction is precise, straightforward, frames the study against the main objectives, and the novelty of the study is properly articulated. The materials and methods section has been presented in such a manner that it can be reproducible by other researchers. The results and discussion section is accompanied with proper discussions on the science behind the results, with appropriate and proper comparison with the results of similar/related studies, and the conclusion properly summarizes the outcome of the study. However, I have the following concerns.

From the introduction, a new application of 1-MCP, which clathrates 1-MCP with α-cyclodextrin was employed, in contrast to the traditional 1-MCP. This should be reflected in the title, as the title suggests the use of the traditional 1-MCP. It is also necessary to reflect this throughout the manuscript wherever 1-MCP is mentioned, especially in the results and discussion section. For example, line 146 could be revised as: “……while all the 1-MCP and α-cyclodextrin clathrate treatment groups retained….” The authors could use “1-MCP and α-cyclodextrin clathrate” or any abbreviation they so choose.

In view of the above, I am also of the opinion that a second control group the utilizes the traditional 1-MCP without α-cyclodextrin clathrate should be introduced for proper comparison. A brief description of the control sheets should also be provided in section 2.1.    

In addition, the authors merely carried out simple analysis, which shows poor scientific sufficiency of the manuscript. The postharvest quality of papaya fruit transcends the few nutritional properties evaluated in the study. It should be all inclusive and bioactives like carotenoids, ascorbic acid, total phenols, total flavonoids and antioxidant capacity/activity should be evaluated. Therefore, additional experiments are required.

Finally, the conclusion should be accompanied by a critical analysis of the impact of the results and the implications for the food industry, followed by relevant recommendations for future study.

Author Response

Comment 1: The manuscript evaluated the effects of humidity-triggered controlled-release 1-methylcyclo-2 propene (1-MCP) clathrates with α-cyclodextrin on postharvest quality of papaya fruit. The manuscript presents an interesting read, abstract is properly written and appropriately reflects the results of the study. The introduction is precise, straightforward, frames the study against the main objectives, and the novelty of the study is properly articulated. The materials and methods section has been presented in such a manner that it can be reproducible by other researchers. The results and discussion section is accompanied with proper discussions on the science behind the results, with appropriate and proper comparison with the results of similar/related studies, and the conclusion properly summarizes the outcome of the study. However, I have the following concerns.

Response 1: Thank you very much for your positive comments. We have revised the manuscript according to your suggestions. Our point-to-point responses are as follows.

Comment 2: From the introduction, a new application of 1-MCP, which clathrates 1-MCP with α-cyclodextrin was employed, in contrast to the traditional 1-MCP. This should be reflected in the title, as the title suggests the use of the traditional 1-MCP. It is also necessary to reflect this throughout the manuscript wherever 1-MCP is mentioned, especially in the results and discussion section. For example, line 146 could be revised as: “……while all the 1-MCP and α-cyclodextrin clathrate treatment groups retained….” The authors could use “1-MCP and α-cyclodextrin clathrate” or any abbreviation they so choose.

Response 2: This study aiming report the papaya edible quality affected by 1-methylcyclopropene clathrate, which however, do not contain the preparation and characterization of the clathrate. So, we describe the sheet based on its characteristics in the manuscript tittle, but we revised it to 1-methylcyclopropene clathrate of α-cyclodextrin (1-MCP@α-cyclodextrin) in the result and discussion section according to your suggestion, and all the revisions have been colored in red.

Comment 3: In view of the above, I am also of the opinion that a second control group the utilizes the traditional 1-MCP without α-cyclodextrin clathrate should be introduced for proper comparison. A brief description of the control sheets should also be provided in section 2.1.

Response 3: We compared the traditional 1-MCP to the 1-MCP@α-cyclodextrin clathrate in our further study. We have supplied some discussion regarding of this. We have also provided a brief description introducing the sheet in section 2.1 according to your suggestion.

Comment 4: In addition, the authors merely carried out simple analysis, which shows poor scientific sufficiency of the manuscript. The postharvest quality of papaya fruit transcends the few nutritional properties evaluated in the study. It should be all inclusive and bioactives like carotenoids, ascorbic acid, total phenols, total flavonoids and antioxidant capacity/activity should be evaluated. Therefore, additional experiments are required.

Response 4: This is our first report studying the effects of humidity-triggered controlled-release 1-MCP on the postharvest quality of papaya. The purpose is to demonstrate if it can be used to extend the storage life of postharvest papaya. So, we emphasized the edible quality. Our further study is working on the physiological and nutrients change in response to the controlled-release 1-MCP.

Comment 5: Finally, the conclusion should be accompanied by a critical analysis of the impact of the results and the implications for the food industry, followed by relevant recommendations for future study.

Response 5: We’ve revised the conclusion (Lines 378-379).

Reviewer 5 Report

This study investigated the effect of humidity-triggered controlled-release 1-methylcyclo- 2 propene (1-MCP) on postharvest quality of papaya fruit. Treating papaya fruit with 1-MCP at 1.0-2.0 ppm preserved its post-harvest quality, resulting in slower ripening and not affecting sensory characteristics. The novelty of this study is that the authors developed an inclusion complex of 1-MCP with α-cyclodextrin, and certain humidity and temperature triggered the release of 1-MCP, solving the problem of controlling its release. However, the following issues persist:

1. Although the authors established various specific humidity levels to simulate different preservation conditions in the study, the release of 1-MCP sheets under different humidity conditions remains unknown.

2. The relationship between 1.0-2.0 ppm of 1-MCP and specific humidity is not explained.

3. The authors employed the size of controlled 1-MCP sheets to address the challenge of controlling the released amount of 1-MCP, yet did not clarify the relationship between the release of 1-MCP from different sizes of 1-MCP sheets  and humidity.

4. There is a lack of studies related to enzymes and nutrients, with the majority of the focus centered on surface-level research.

5. Sensory studies should have been placed earlier in the article, prioritizing them as a fundamental component of vegetable assessment.

6. Some grammatical errors exist, and many references are outdated.

Suggestions:

1.Research the influence of varying humidity levels on 1-MCP release and explore additional membrane indicators to highlight the performance of controlled release tablets.

2.Elaborate on the correlation between 1.0-2.0 ppm of 1-MCP and specific humidity to enhance the logical coherence of the article.

3.Provide an explanation for the relationship between the release of 1-MCP and humidity concerning different sizes of 1-MCP sheets.

4.Incorporate relevant enzyme and nutrient-related studies to enhance the article's depth.

5.Consider merging Table 2 and Table 5 for more comprehensive analysis.

6.Include references from the last five years, particularly experimental literature, to provide more recent and relevant sources.

Moderate editing of English language required

Author Response

Comment 1: This study investigated the effect of humidity-triggered controlled-release 1-methylcyclo- 2 propene (1-MCP) on postharvest quality of papaya fruit. Treating papaya fruit with 1-MCP at 1.0-2.0 ppm preserved its post-harvest quality, resulting in slower ripening and not affecting sensory characteristics. The novelty of this study is that the authors developed an inclusion complex of 1-MCP with α-cyclodextrin, and certain humidity and temperature triggered the release of 1-MCP, solving the problem of controlling its release. However, the following issues persist:

Response 1: Thank you for the positive comments. We have revised the manuscript according to your suggestions. Our point-to-point responses are as follows.

Comment 2: Although the authors established various specific humidity levels to simulate different preservation conditions in the study, the release of 1-MCP sheets under different humidity conditions remains unknown.

Response 2: The 1-MCP sheet was cut into different sizes according to the manufacturer's guidance, and it was calculated based on the product’s total 1-MCP amount and the release rate. The theoretical release rate, dosage, and actual concentration can be calculated from the manufacturer's patent.

Reference:

Wood, W; Sarageno, J.F; Keute, J; Lundgren A. Compositions and methods for differential release of 1-methylcyclopropene. US Patent 11492419B2, filed 17 February 2021, issued 8 November 2022.

Comment 3: The relationship between 1.0-2.0 ppm of 1-MCP and specific humidity is not explained.

Response 3: According to the manufacturer's guidance, 1-MCP will release from the sheet when the humidity is over 80%. The 1-MCP dosage is positively linear correlated with the sheet size.

Comment 4: The authors employed the size of controlled 1-MCP sheets to address the challenge of controlling the released amount of 1-MCP, yet did not clarify the relationship between the release of 1-MCP from different sizes of 1-MCP sheets and humidity.

Response 4: The 1-MCP sheet was cut into different sizes according to the manufacturer's guidance, and it was calculated based on the product’s total 1-MCP amount, the release rate, and the volume of the fruit. Since the fruit were stored under the constant temperature and humidity conditions, the 1-MCP release rate was only affected by the sheet’s size.

Comment 5: There is a lack of studies related to enzymes and nutrients, with the majority of the focus centered on surface-level research.

Response 5: This is our first report studying the effects of humidity-triggered controlled-release 1-MCP on the postharvest quality of papaya. The purpose is to demonstrate if it can be used to extend the storage life of postharvest papaya. So, we emphasized the edible quality. Our further study is working on the physiological and nutrients change in response to the controlled-release 1-MCP.

Comment 6: Sensory studies should have been placed earlier in the article, prioritizing them as a fundamental component of vegetable assessment.

Response 6: The sensory study was conducted to simulate the fruit after transit and store at room temperature for 10 days. This section was to demonstrate after the cold storage and shelf-life, the treated fruit was still in the edible stage, along with the sensory evaluation conducted on the last day of the experiment, therefore we prefer it should be on the last of the manuscript.

Comment 7: Some grammatical errors exist, and many references are outdated.

Response 7: We have revised the manuscript to correct grammatical errors, and the changes have been marked in red. At the same time, we have supplemented research of the last 5 years to follow up on recent research developments.

Comment 8: Research the influence of varying humidity levels on 1-MCP release and explore additional membrane indicators to highlight the performance of controlled release tablets.

Response 8: According to the manufacturer's guidance, 1-MCP will release from the sheet when the humidity is over 80%. The different 1-MCP dosage is based on the different sheet size, and the release rate is positively correlated with temperature. We also supplied some discussion to show the importance of fruit tissue membrane integrity, and those will be in our future study.

Comment 9: Elaborate on the correlation between 1.0-2.0 ppm of 1-MCP and specific humidity to enhance the logical coherence of the article. Provide an explanation for the relationship between the release of 1-MCP and humidity concerning different sizes of 1-MCP sheets.

Response 9: According to the manufacturer's guidance, the release rate is affected by the particle size of the 1-MCP and α-cyclodextrin clathrate. The 1-MCP gas complexes readily with α-cyclodextrin to form a crystalline solid that is easily collected as a powder. The powder was printed on the sheet, comprising a composition consisting 1-MCP clathrate of α-cyclodextrin (1-MCP@α-cyclodextrin) and its mean particle size was between 3 μm and 15 μm. The sheet was positioned on the fruit, where the humidity of biological respiration caused 1-MCP disgorgement. Disgorgement conditions refer to the atmospheric conditions of ambient pressure (about 1 atm), temperature between 0° C and about 50° C., and relative humidity of above 80%

Comment 10: Incorporate relevant enzyme and nutrient-related studies to enhance the article's depth.

Response 10: This is our first report studying the effects of humidity-triggered controlled-release 1-MCP on the postharvest quality of papaya. The purpose is to demonstrate if it can be used to extend the storage life of postharvest papaya. So, we emphasized the edible quality. Our further study is working on the physiological and nutrients change in response to the controlled-release 1-MCP.

Comment 11: Consider merging Table 2 and Table 5 for more comprehensive analysis.

Response 11: Thank you for your suggestion. However, Figure 5 only contains one timepoint data, which cannot be merged with other tables or figures.

Comment 12: Include references from the last five years, particularly experimental literature, to provide more recent and relevant sources.

Response 12: We have revised the manuscript to correct grammatical errors, and the changes have been marked in red. At the same time, we have supplemented research of the last 5 years to follow up on recent research developments.

Round 2

Reviewer 3 Report

The authors have answered all questions and modified almost issues. There is still a mistake needed to be corrected. In the second row of Table 1, 16+8 and 16+10 should be changed to 14+8 and 14+10.

Author Response

The authors have answered all questions and modified almost issues. There is still a mistake needed to be corrected. In the second row of Table 1, 16+8 and 16+10 should be changed to 14+8 and 14+10.

Thank you for your comments. 16+8 and 16+10 have been changed to 14+8 and 14+10. 

Reviewer 4 Report

The authors have blatantly refused to make corrections based on my suggestions and have not given reasons why they refused to carry out the suggestions.

For starters, I requested the clathrate of 1-MCP with α-cyclodextrin to be reflected in the title, and everywhere in the manuscript where it is referred to. This was not done.

I also suggested a second control group that utilizes the traditional 1-MCP without α-cyclodextrin clathrate for proper comparison. This was not done.

I further stated that the authors merely carried out simple analysis, which shows poor scientific sufficiency of the manuscript, and requested additional properties like carotenoids, ascorbic acid, total phenols, total flavonoids and antioxidant capacity/activity to be studied. The authors have refused to do this, neither have they provided reasons for not effecting the corrections.

Minor editing of language is required 

Author Response

The authors have blatantly refused to make corrections based on my suggestions and have not given reasons why they refused to carry out the suggestions.

For starters, I requested the clathrate of 1-MCP with α-cyclodextrin to be reflected in the title, and everywhere in the manuscript where it is referred to. This was not done.

Thank you for your suggestion. We feel including ‘humidity-triggered controlled-release’ in the title sufficiently indicates that we are not using traditional 1-MCP generation.

I also suggested a second control group that utilizes the traditional 1-MCP without α-cyclodextrin clathrate for proper comparison. This was not done.

Since the 1-MCP in the sheet is controlled-release, and the concentration of 1-MCP is positively linearly correlated with the sheet size under the constant temperature and relative humidity, and the carrier α-cyclodextrin has no impact on fruit quality, control groups with the same concentrations (0.5 ppm, 1.0 ppm, 2.0 ppm, or 4.0 ppm) using traditional 1-MCP generation are not necessary.

I further stated that the authors merely carried out simple analysis, which shows poor scientific sufficiency of the manuscript, and requested additional properties like carotenoids, ascorbic acid, total phenols, total flavonoids and antioxidant capacity/activity to be studied. The authors have refused to do this, neither have they provided reasons for not effecting the corrections.

The main focus of current study is to evaluate the effect of 1-MCP on physical (color, firmness, weight loss), chemical (sugar, acid), microbial (decay rate), and sensory (taste) properties of papaya, to make sure the treated fruit are edible and marketable. The nutritional properties (carotenoids, ascorbic acid, total phenols, total flavonoids and antioxidant capacity) you suggested will be good for a future study. Thank you.

Reviewer 5 Report

accept

ok

Author Response

accept

Thank you!